# EDGE-SAMPLER: EFFICIENT IMPORTANCE SAMPLING FOR NEURAL IMPLICIT SURFACES RECONSTRUCTION

## ABSTRACT

Neural implicit surfaces have attracted much attention in the 3D reconstruction field. Equipped with signed distance functions (SDFs), neural implicit surfaces significantly improve geometry reconstruction quality compared to neural radiance fields (NeRFs). However, compared with NeRFs, training SDFs is more challenging and time-consuming because it requires large sample counts to sample the thin edges of implicit surface density functions. Up till today, error-bounded sampling is the sole volume importance sampling technique dedicated to implicit SDFs, which theoretically bounds the errors of sample weights and thus prevents missing important thin surface edges, but at the cost of large sample counts. In this work, we introduce an efficient edge-sampler technique to significantly reduce the required sample counts by up to 10x while still preserving the theoretical error bound by reducing Riemann integral bias. Specifically, the technique first proposes a double-sampling strategy to detect the thin intervals of surface edges containing all valid samples. Then, it fits the density functions of the intervals with bounded cumulated distribution functions (CDF) errors and produces the final Riemann sum with sparse uniform samples. Extensive results in various scenes demonstrate the superiority of our sampling technique, including improving geometry reconstruction details, significantly reducing sample counts and training time, and the capability to be generalized to various implicit SDF frameworks.

## 1 INTRODUCTION

3D reconstruction from multi-view images is a classic research area in both computer vision and computer graphics. Recently, neural radiance field (NeRF) (Mildenhall et al., 2020) and its subsequent works have demonstrated great potential for static or dynamic scene reconstruction. However, due to the lack of effective constraints on geometry, the reconstructed geometry of NeRFs generally suffers from discernible noise and artifacts. Neural implicit surfaces (Yariv et al., 2021; Wang et al., 2021) significantly improve the reconstruction results by constraining the scene to SDF fields, thus achieving SOTA geometry reconstruction quality. However, the implicit SDF reconstruction requires hours to train multi-layer perceptron (MLP) networks via dense sampling algorithms, limiting its applications in practice. Some acceleration methods have been proposed to cope with this problem, which mainly focuses on two aspects: sophisticated spatial coding algorithms for reducing network parameters and accurate sampling for faster convergence speed. While the adaption of several effective spatial coding algorithms (Takikawa et al., 2021; Barron et al., 2021; Müller et al., 2022) to SDF have produced promising results, fast sampling algorithms dedicated to neural implicit surfaces are still not well-studied.

Representatively, VolSDF (Yariv et al., 2021) proposed an error-bounded sampling method dedicated to Laplace distribution-based neural implicit surfaces by mapping the SDF values to density values. They first derived the maximum Riemann sum error of the weights along rays, and iteratively increased the number of sampling points until the maximum error was less than a preset value. As long as the sampling is dense enough, weights estimated using the SDF values of the sampling points are close to the true value. Then, it utilizes the weights as PDFs to sample points for training the SDF. Thanks to the error-bounded sampling strategy, this method converges fast and will not miss thin surface edges. However, this approach has two major drawbacks. First, it requires plenty of sampling points that need to query MLP to obtain SDF values. The typical number of MLP queries per ray is over 700, making the sampling process relatively slow. Second, their method can-

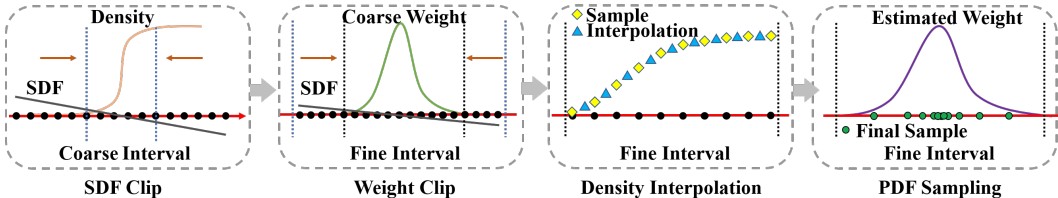

Figure 1: The overview of our edge-sampler algorithm. First, a double-sampling strategy (SDF clip and weight clip) is designed to determine the valid compact interval. Then, the maximum error of integrated weights is calculated to guide the interpolation of the density of interval sparse sampling points (density interpolation). Finally, estimate the weight function as sampling PDF with all density values, and perform final inverse sampling of CDF (PDF sampling).

not handle neural implicit surfaces for non-Laplace distribution-based mapping like NeuS (Wang et al., 2021).

To overcome these drawbacks, we propose *edge sampler*, a novel sampling method for fast training neural implicit surfaces, see Figure 1. Our method is designed based on three objectives: **(1) Accuracy.** Whether neural radiance fields or neural implicit surfaces, the distribution of the final sampling points on a ray should conform to the weight function as much as possible. To achieve this goal, VolSDF densely samples the ray to calculate the maximum weight errors and then integrates the errors to for sampling density control. However, applying Riemann integration to the weights error creates a biased and over-conservative weight error bound, as we demonstrate in Section 4.3. Instead, we propose to calculate an error bound on the integration of weights, leading to a tighter error bound and producing better sample distributions. **(2) Efficiency.** We first design a coarse-to-fine sampling strategy called double-sampling to calculate the valid sampling interval for each ray expeditiously, which aims to include all valid weights in an interval as small as possible. Then, only a few queries of uniform sampling points that are distributed in the sampling interval can obtain accurate weights. Different from (Mildenhall et al., 2020), our coarse-to-fine strategy does not need to train a coarse network and can be directly calculated from the sampling interval. **(3) Universality.** Our method is applicable to all neural implicit surfaces with equations mapping SDF values to density values under the volume rendering framework. Specifically, we evaluate common mapping equations, including the Laplace distribution and the logistic distribution in practice.

Through experiments, we illustrate that our method can achieve better geometry reconstruction results while accelerating the training of neural implicit surfaces. Specifically, compared with VolSDF, our sampling method achieves an acceleration of more than 10X at most. The total training time of scenes on DTU dataset (Aanæs et al., 2016) is reduced by 60% at most compared to VolSDF with hash encoding, and 79% at most compared to original VolSDF. In the qualitative results, our reconstructed meshes acquire richer details compared to proposal sampling methods and occupancy grid sampling methods. The main contributions of our work are summarized as follows:

• We propose *edge sampler*, a fast, accurate, and generalized sampling method for high fidelity neural implicit surfaces models training acceleration.

• We derive the maximum error of integrated weights to guide the sampling process, which ensures the accuracy of our sampling method.

• We propose several novel training strategies, including a double-sampling strategy for fast sampling interval calculation, piecewise fitting and Gaussian Process strategies for density interpolation, and an error-guided ray sampling strategy for improving color reconstruction quality.

## 2 RELATED WORK

**Neural implicit surfaces.** VolSDF (Yariv et al., 2021) and NeuS (Wang et al., 2021) are two classic methods of neural implicit surfaces which both have a series of follow-up work. They are both designed under NeRF's volume rendering framework, but VolSDF applies Laplace distribution mapping SDF values to density values, while NeuS uses logistic distribution. Meanwhile, the density mapping method of NeuS is unbiased in planar scenes, compared with VolSDF. (Fu et al., 2022; Yu et al., 2022) improve reconstruction by introducing geometric cues. (Ge et al., 2023) focus on

reconstructing objects with strong reflection. (Fan et al., 2023) not only achieves the reconstruction of glossy objects, it also estimates the illumination and material. Similarly, (Yariv et al., 2023) also reconstructs the material with geometry and even achieves real-time rendering. (Jiang et al., 2023; Azinović et al., 2022) replace RGB inputs with RGB-D inputs which naturally results in higher reconstruction quality. In order to speed up training, (Rosu & Behnke, 2023) replaces the voxel hash encoding with a permutohedral lattice which optimizes faster. Recently, (Wang et al., 2023) deeply combining NeuS with instant-ngp (Müller et al., 2022) which also significantly reduces the training time of original NeuS. In the popular AIGC field, implicit SDFs can also generate 3D content (Xu et al., 2023; Zheng et al., 2022).

**Sampling methods.** In general, sampling methods for neural implicit surfaces can be classified into 4 categories: error-bounded (VolSDF) method, coarse-to-fine method, voxel-surface guided method, sampling network method, and occupancy grid method. Except error-bounded method which has already introduced above, the rest are sampling algorithms derived from NeRFs. The coarse-to-fine method (Mildenhall et al., 2020) guides sampling by training a coarse network. They sample uniformly on the coarse network and consider the density value of the sampling point as PDF to guide sampling on the fine network. This sampling method is inefficient due to training an additional network. Occupancy grid method (Li et al., 2022; Müller et al., 2022) discretizes the scene into voxels, and query the grid to determine whether the sampling point contributes to the color of rays, thereby skipping invalid areas and achieving sampling acceleration. However, the accuracy of the query depends on the grid resolution. High-resolution grid takes up extra GPU memory. Also, occupancy grid requires a large number of MLP queries when updating, and low update frequency greatly reduces query accuracy. Voxel-surface guided method (Sun et al., 2022) is an efficient sampling algorithm which combines occupancy grid with surface guided sampling. However, their method requires pre-reconstructed point clouds, which limits the applicable scenarios of the algorithm. Sampling network methods (Lindell et al., 2021; Piala & Clark, 2021; Barron et al., 2022; Kurz et al., 2022) train neural networks to directly sample or guide sampling alone rays with end-to-end or pre-training manners. For end-to-end training methods, e.g. proposal networks, gradients calculation and backpropagation consume additional time. For pre-trained methods, network predictions for unknown scenes are unreliable.

## 3 PRELIMINARIES

Neural implicit surfaces share the same framework with NeRF (Mildenhall et al., 2020), which represents a scene by a neural network (usually a MLP). For any ray parametrized as $\mathbf{r}(t) = \mathbf{o} + t\mathbf{d}$ passing through the scene, the classic volume rendering equation takes density $\sigma$ and color $\mathbf{c}$ of samples predicted by the neural network to provide a solution for ray color:

$$C(\mathbf{r}) = \int_{t_n}^{t_f} T(t)\sigma(\mathbf{r}(t))\mathbf{c}(\mathbf{r}(t), \mathbf{d})dt, \text{ where } T(t) = \exp\left(-\int_{t_n}^{t} \sigma(\mathbf{r}(s))ds\right), \quad (1)$$

where $C(\mathbf{r})$ denotes the predicted color of the ray, $\sigma(\mathbf{r}(t))$ denotes the volume density of point $\mathbf{r}(t)$, $t_n$ and $t_f$ are the near and far sampling distances, $T(t)$ denotes the accumulated transmittance along the ray from $t_n$ to $t$. However, directly predicts the density of samples lacking geometric constraints. Therefore, neural implicit surfaces utilize the network to predict SDF values of samples and map SDF values to density values through the mapping function. CDF of the Laplace distribution with zero means and $\beta$ scale is one of the commonly used equations proposed by VolSDF (Yariv et al., 2021):

$$\sigma_L(\mathbf{r}(t)) = \alpha\Psi_\beta\left(-d_\Omega(\mathbf{r}(t))\right), \text{ where } \alpha = \left(\frac{1}{\beta}\right), \quad (2)$$

$$\Psi_\beta(s) = \begin{cases} \frac{1}{2}\exp\left(\frac{s}{\beta}\right) & \text{if } s \leq 0 \\ 1 - \frac{1}{2}\exp\left(-\frac{s}{\beta}\right) & \text{if } s > 0 \end{cases}, \quad (3)$$

where $d_\Omega$ denotes the predicted SDF value of a sampling point. $\beta$ is a learnable parameter that approaches zero during training. However, this mapping function is biased because $\Psi_\beta$ does not reach a global maximum where the SDF value is zero, i.e., sampling points on the surface. Therefore, NeuS (Wang et al., 2021) proposed an unbiased mapping function — logistic distribution:

$$\sigma_l(d_\Omega(\mathbf{r}(t))) = se^{-sd_\Omega(\mathbf{r}(t))}/\left(1 + e^{-sd_\Omega(\mathbf{r}(t))}\right)^2, \quad (4)$$

where $s$ is the reciprocal of standard deviation of $\sigma$, which also approaches to zero during training as a trainable parameter. This mapping function cannot be applied directly because it is not occlusion-aware. Under the assumption of a planar scene, the final mapping equation is derived as follows:

$$\sigma_l(d_\Omega(\mathbf{r}(t))) = \max\left(\frac{-\frac{\mathrm{d}\Phi_s}{\mathrm{d}t}(d_\Omega(\mathbf{r}(t)))}{\Phi_s(d_\Omega(\mathbf{r}(t)))}, 0\right), \tag{5}$$

where $\Phi_s$ denotes the Sigmoid function, i.e., $\sigma(x) = \Phi'(x)$.

In practice, the continuous function integral in Equation 1 is converted into a discrete Riemann sum. $N$ points $\{\mathbf{p}_i = \mathbf{o} + t_i\mathbf{d} \mid i = 1, \ldots, N\}$ is sampled along the ray to approximate pixel color:

$$\hat{C}(\mathbf{r}) = \sum_{i=1}^{N} T_i\left(1 - \exp\left(-\sigma_i\delta_i\right)\right)\mathbf{c_i}, \text{ where } T_i = \exp\left(-\sum_{j=1}^{i-1}\sigma_j\delta_j\right), \tag{6}$$

$\delta_i$ denotes the length of interval $s_i$ from $\delta_i$ to $\delta_{i+1}$, $\alpha_i = (1 - \exp(-\sigma_i\delta_i))$ is the alpha value of interval $s_i$, $T_i\alpha_i$ is called the weight of a sampling point by convention, denoted by $\omega_i$. It is obvious that the expected ray color is estimated by $\omega_i$, and thus the optimal sampling distribution should be proportional to $\omega_i$. Neural implicit surfaces consider $\{\omega_i \mid i = 1, \ldots, N\}$ as a piecewise function of sampling PDF:

$$PDF(x) = \frac{\omega_i}{\sum_{i=1}^{N}\omega_i}, \qquad x \in [t_i, t_{i+1}). \tag{7}$$

The CDF obtained by integrating this PDF is used as the final inverse sampling.

To optimize the network, the color loss $\mathcal{L}_{\text{color}}$ is defined as:

$$\mathcal{L}_{\text{color}} = \sum_{\mathbf{r} \in \mathcal{R}} \|C'(\mathbf{r}) - C(\mathbf{r})\|_1, \tag{8}$$

where $\mathcal{R}$ denotes a batch of training rays.

## 4 METHOD

### 4.1 OVERVIEW

Given an arbitrary ray passing through the scene, our goal is to accurately estimate the weight distribution of the ray with fairly low computational cost, making the distribution of sampling points and the sample weights as close as possible. This is a difficult task because SDF density functions converge to an indicator function, which means valid weights are distributed in a tiny interval very close to the surface. Without dense sampling, tiny intervals can easily be missed, which leads to inaccurate estimation of equation 6 and loss gradient. To cope with this challenge, we designed a four-step sampling technique, as shown in Figure 1. Each step is discussed in detail in the following sections. Besides, a novel ray sampling strategy for improving color reconstruction quality is also mentioned.

### 4.2 DOUBLE-SAMPLING STRATEGY

The key idea of the double-sampling strategy is to replace the dense sampling used to find the tiny edge interval with two passes of sparse samplings and thus reduce the calculation cost while ensuring accuracy. First, we sparsely sample a given ray over its near and far ends and evaluate the SDF values of these sampling points by querying the MLP. The sampling points are then clipped based on the **valid SDF bound** as formerly defined later. Benefiting from the mapping function, we can analytically calculate an SDF bound that satisfies density equals to a small constant $\epsilon$, which provides a new direction for dedicated neural implicit surfaces sampling. Due to the monotonicity of the mapping function, the density of any SDF greater than this value will be less than this $\epsilon$, which can be ignored. Thus skipping meaningless areas. Here we take Laplace distribution as an example to calculate the SDF bound:

$$SDF_{bd} = |\beta\log(2\epsilon)|. \tag{9}$$

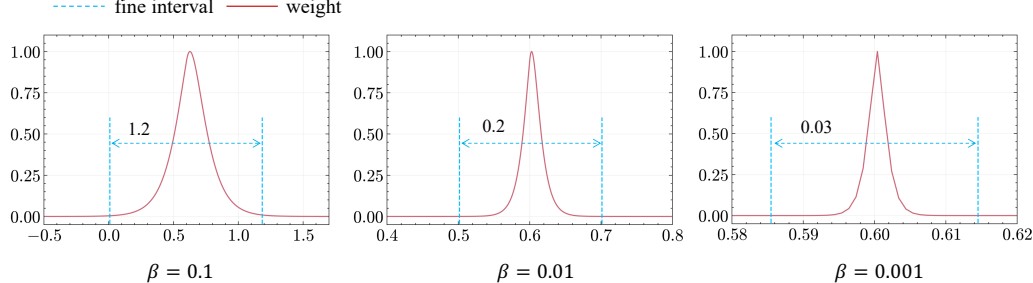

Figure 2: Weights and $interval_f$ of a ray under different $\beta$. As $\beta$ decreases, $interval_f$ always surrounds the valid weights. Notice that the length of $interval_f$ is only 0.03 when $\beta$ is 0.001, while the ray length is 6.

During training, $SDF_{bd}$ decreases as $\beta$ decreases, which is getting closer to the surface. To avoid skipping the thin surface edge, we first traverse all sampling points forward, find the first sampling point $sample_i$ whose SDF value is smaller than $SDF_{db}$, and discard $sample_1$ to $sample_{i-2}$ (included), which determines the lower bound $sample_l$. If SDF values of all sampling points are greater than $SDF_{db}$, no samples will be discarded. Then, sampling points are traversed inverse with a similar operation which determines the upper bound $sample_u$. $sample_l$ and $sample_u$ determine the coarse interval $interval_c$, see SDF Clip in Figure 1.

Equipped with $interval_c$, we can skip most of the zero-valued sampling intervals. However, this interval is not small enough to closely surround surface edge, especially when $\beta$ (in equation 3) or $s$ (in equation 4) is small, $T$ (in equation 1) quickly converges to zero, although the density value is still valid. Thus, We perform another sparse sampling in the $interval_c$, and query their SDF values to calculate the valid **weight bound** to further clip the interval. The coarse weights are calculated using these SDF values, and the **weight bound** is determined by the maximum value of coarse weights. Then, approaches similar to SDF Clip are applied to find the fine interval $interval_f$. Please refer to Weight Clip in Figure 1.

The $interval_f$ specifies the compact interval of the surface edge. During training, it can be reduced to **one over two hundred** of the total ray sampling interval as illustrated in Figure 2.

### 4.3 ERROR-BOUNDED WEIGHT SAMPLING

Given that $interval_f$ is compact enough, we uniformly distribute sparse samples to fit a PDF under a bounded error. Different from the error-bounded sampling method that only bounds the error of weight of individual points, we prove that our method can bound the accumulated weight error to any small value.

**Riemann sum error estimation.** Suppose we sample $N$ points, $sample_1$ to $sample_N$, through $interval_f$, the distance between each $interval_i$ ($sample_i$ and $sample_{i+1}$) is $dis_i$ $\omega_i$ denotes the corresponding weight of each sample and $\omega_1 = 0$. The estimate the (left) Riemann sum of weight function $\Omega$ in $interval_f$ is:

$$Rei(\Omega) = \sum_{i=2}^{N} \omega_i dis_{i-1}. \qquad (10)$$

Now we consider the error between the Riemann sum and the integral on each $interval_i$. Assume that $\Omega$ is monotonically increasing within $interval_f$, as shown in Figure 3, the Riemann sum value is always greater than the true integral value. Due to the monotonicity of

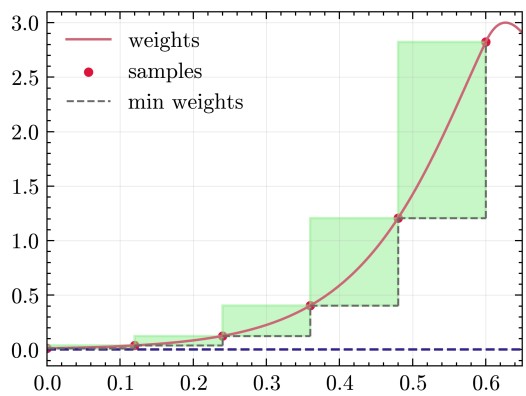

Figure 3: Schematic diagram of the Riemann sum error. For a monotonically increasing interval, the Riemann sum is always greater than the true integral value.

$\Omega$, the minimum value within each $interval_i$ is $\omega_i$ (the gray dashed line). Therefore, the maximum error in each $interval_i$ is $(\omega_i - \omega_{i-1})\,dis_{i-1}$ (the green square). The total max error through $interval_f$ is:

$$ErrorMax_{increase} = \sum_{i=2}^{N} (\omega_i - \omega_{i-1})\,dis_{i-1} \tag{11}$$

$$= \sum_{i=2}^{N-1} \omega_i\,(dis_{i-1} - dis_i) + \omega_n dis_{n-1} - \omega_1 dis_1. \tag{12}$$

While applying a uniform sampling to estimate the final integration, in which all $dis_i$ equal to $dis_u$, equation 12 can be simplified to a simple formula:

$$ErrorMax_{increase} = \omega_n dis_u, \tag{13}$$

where $dis_u$ is the length of each interval. Obviously, for a monotonically increasing function, the max error is positive and also monotonically increasing. With a similar approach, we can also prove that for a monotonically decreasing function, the max error is negative and monotonically decreasing:

$$ErrorMax_{decrease} = -(\omega_1 - \omega_n)dis_u. \tag{14}$$

Whether the weight functions are based on Laplace distribution or logistic distribution, they are all bell-shaped functions, i.e., first monotonically increase to reach the maximum value and then decrease monotonically to zero. Therefore, we divide it into two monotonic functions and analyze them separately. Assume that the weight function takes the maximum value $\Omega_{max}$ at $pos_t$, then the absolute values of $ErrorMax_{increase}$ and $ErrorMax_{decrease}$ are all equal to $\Omega_{max}dis_u$ due to equation 13 and equation 14.

**Error bound of integrated weights.** Suppose that the true integral value of the whole weight function is $\mathbf{W}$, then the maximum and minimum value of the Riemann sum is $\mathbf{W} + \Omega_{max}dis_u$ and $\mathbf{W} - \Omega_{max}dis_u$. Meanwhile, the maximum error is $\Omega_{max}dis_u$ at $pos_t$.

Multiply the numerator and denominator of the equation 7 by $dis_u$, the PDF and CDF function is converted into Riemann sum form:

$$PDF(x) = \frac{\omega_i dis_u}{\sum_{i=1}^{N} \omega_i dis_u}, \qquad x \in [t_i, t_{i+1}). \tag{15}$$

$$CDF(x) = \sum PDF(x) \tag{16}$$

$$= \frac{\sum_{j=1}^{i} \omega_j dis_u}{\sum_{i=1}^{N} \omega_i dis_u}, \qquad x \in [t_i, t_{i+1}). \tag{17}$$

Obviously, the denominator of CDF is the Riemann sum of $\Omega$, and the numerator is the Riemann sum up to $sample_i$. According to the above analysis, the maximum error value of the numerator is $\Omega_m axdis_u$, and the minimum value of the denominator is $\mathbf{W} - \Omega_{max}dis_u$. Thus, **the error bound of integrated weights** is:

$$ErrorMax(dis_u) = \frac{\Omega_{max}dis_u}{\mathbf{W} - \Omega_{max}dis_u}, \tag{18}$$

which is a monotonically increasing function of $dis_u$. When $dis_u$ is small enough, that is, the sampling is dense enough, it converges to zero. In practice, $\Omega_{max}$ is hard to get, so we approximate it by the maximum value $\omega_{max}$ among $\omega_i$.

However, the above analysis is based on the accuracy of $\omega_i$, which is also **biased for Laplace distribution** due to equation 6, where $T_i$ is determined by the Riemann sum of density $\sigma_i$. Therefore, we analyze the error bound of weights in a similar manner. Density functions (equation 3 and equation5) are all monotonically increasing functions, thus assuming the true density interval up to $density_i$ is $\mathbf{D_i}$, the max error can be directly derived from equation 13:

$$ErrorMax_{T_i} = \exp(-\mathbf{D_i}) - \exp(-(\mathbf{D_i} + \sigma_i dis_u)). \tag{19}$$

When $dis_u$ is small enough, it converges to zero. Notice that equation 19 is always positive, which means $weight_{max}$ is smaller than the true value. Therefore, the weight bias should be taken into account when calculating equation 18. For logistic distribution, $\omega_i$ is obtained by analytically solving the integral of $T$ in equation 1. Therefore, only the error of integrated weights need to be considered.

## 4.4 SAMPLES INTERPOLATION

Now we can estimate the Riemann sum error for any uniform sampling through $interval_f$. Assume that $N$ points are sampled, the corresponding interval length is $dis_n$, and the Riemann sum of density function up to $sample_i$ is $RieD_i$. For Laplace distribution, we first calculate the max error of $\omega_i$ by equation 6 and 19:

$$ErrorMax_{weight} = \max_{i \in [N]}(|\alpha_i(\exp(-RieD_i) - \exp(-(RieD_i - \sigma_i dis_n)))|). \tag{20}$$

Then we calculate $\omega_i$ by equation 6 and modify the $\omega_{max}$ with $ErrorMax_{weight}$: $\omega_{max} = \omega_{max} + ErrorMax_{weight}$. Finally, we substitute $\omega_{max}$ to equation 18 and get the max integrated weights error equation subject to $dis_n$:

$$ErrorMax(dis_n) = \frac{\omega_{max} dis_n}{RieW - \omega_{max} dis_n}, \tag{21}$$

where $RieW$ is the Riemann sum of the entire $\Omega$. For logistic distribution, $\omega_{max}$ is not modified, and equation 21 also holds true. Set equation 21 equals to a tiny constant $\epsilon$, $dis_n$ can be solved, which means $N$ is determined. However, directly sampling $N$ samples in $interval_f$ is still time-consuming, since $N$ is usually greater than 100 during training. Thus, we present two approaches to interpolate densities from sparse sampling. We interpolate the density function because $T_i$ is determined by the Riemann sum of density.

Piecewise fitting (linear interpolation) is a direct and efficient method for interpolation. Through experiments, sampling 16 points can usually ensure the accuracy of the interpolation result. However, the interpolation result is not smooth enough when the derivative of the function is relatively large. Gaussian Process (GP) fitting can handle complex functions that have multiple extreme points with the same 16 sampling points. Their interpolation results are smoother and more accurate than linear interpolation. However, the computational complexity of GP fitting is greatly increased compared to linear interpolation due to the operation of inverting a huge matrix. Please refer to the ablation study section for a detailed analysis.

## 4.5 ERROR SAMPLING

Although neural implicit surfaces mainly focus on geometry reconstruction, image synthesis is also an important application. Due to strong geometric constraints, the quality of image synthesis is not comparable to NeRFs. To address this problem, we propose an important ray sampling method to enhance the image synthesis ability. In each iteration, we sort the RGB errors of all iterated rays in descending order and select the top 1024 rays. These rays are distributed in different pictures and represent the most difficult samples for the current network to predict so far, which are sent to the next iteration as additional rays. Compared with maintaining an error map for each image, our method is more memory-saving. At the same time, the rays for each iteration come from different images, which pay more attention to the global error. The disadvantage of this method is that it will slow down the training, and the geometry reconstruction may become worse. Therefore, error sampling is only turned on during image synthesis tasks.

# 5 EXPERIMENTS

## 5.1 EXPERIMENTS SETUP

We use the DTU Jensen et al. (2014) dataset, which contains multi-view images of different objects and chamfer distance to evaluate our method. We sample 64 or 128 points in the double-sampling stage (32 or 64 for a single pass) and additionally sample 16 points for interpolation. The final samples of our method include 16 CDF inverse samples and 32 uniform samples which will decay to 16 during training. The batch size of all experiments is 8, and the number of sampling rays in each batch is 1024. For the Laplace distribution, we compare the reconstruction quality and training speed with the original VolSDF, i.e., error-bounded sampling (EB), as well as the VolSDF with hash encoding (EB **w/** Hash), and the SDF grid method which is integrated into our sampling framework (Grid32) to replace MLP queries. For logistic distribution, we compare to the original NeuS, and NeuS with hash encoding (NeuS **w/** Hash) methods.

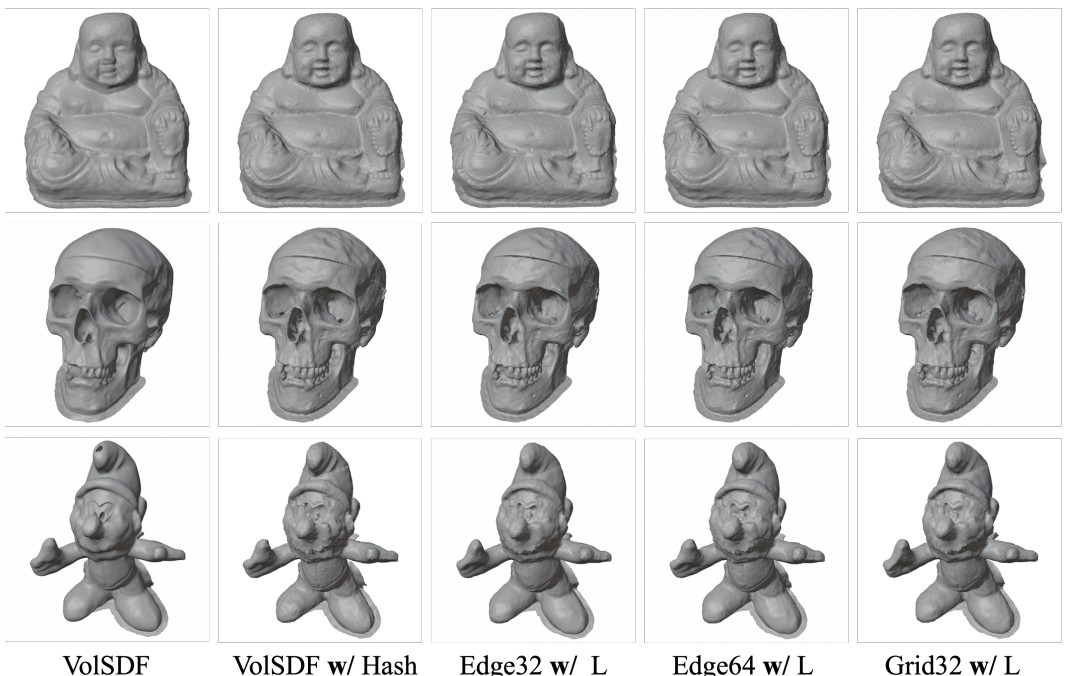

| VolSDF | VolSDF **w/** Hash | Edge32 **w/** L | Edge64 **w/** L | Grid32 **w/** L |

Figure 4:

## 5.2 EVALUATION

**Sampling time comparison.** We compare our sampling time of 8192 rays with various methods in Table 1. "L" and "GP" denote linear interpolation and Gaussian process. "Proposal" is a network sampling method proposed by MipNeRF360 Barron et al. (2022). We also implement our sampling method on logistic distribution (Edge32 **w/**L for NeuS). Our Edge32 **w/** L method is 10 times faster compared to EB, meanwhile, our Edge32 **w/**L for NeuS method also achieves a faster sampling speed compared to original NeuS.

Table 1: Sampling time for multiple methods in seconds.

| EB | Edge32 **w/** L | Edge64 **w/** L | Edge32 **w/** GP | Edge32 |
|---|---|---|---|---|
| 0.335 | 0.031 | 0.052 | 0.058 | 0.029 |
| Grid32 (Sample) | Grid32 (Update) | Proposal | NeuS | Edge32 **w/** L (NeuS) |
| 0.060 | 0.089 | 0.104 | 0.029 | 0.024 |

**Comparison based on Laplace distribution methods.** Our sampling method is not only efficient, but also maintains high-quality geometric reconstruction. Here we compare our method with EB, SDF grid methods, which are all based on Laplace distribution. As shown in Table 2, our Edge32 **w/** L method has the shortest training time and considerable reconstruction quality.

Table 2: Comparison of our Laplace distribution-based sampling method with other methods. We use chamfer distance to measure the accuracy of geometry reconstruction. Our methods achieve the lowest chamfer distance in each scene which represents the best reconstruction quality.

| | EB | | EB **w/** Hash | | Edge32 **w/** L | | Edge64 **w/** L | | Grid32 **w/** L | |
|---|---|---|---|---|---|---|---|---|---|---|
| | Chamfer | Time | Chamfer | Time | Chamfer | Time | Chamfer | Time | Chamfer | Time |
| Scan65 | 1.26 | 22673 | 0.87 | 12684 | **0.84** | **6130** | 0.90 | 6284 | 0.82 | 6309 |
| Scan83 | 1.54 | 27162 | 1.36 | 14763 | 1.48 | **7550** | **1.32** | 7844 | 1.49 | 7913 |
| Scan106 | 0.81 | 27793 | 0.64 | 13951 | **0.58** | **7329** | 0.64 | 7695 | 0.83 | 8132 |
| Scan114 | 0.70 | 27568 | 0.64 | 14482 | **0.46** | **5804** | 0.55 | 7626 | 0.64 | 7708 |

**Comparison based on logistic distribution methods.** Here we compare our method with Neus, Neus **w/** hash encoding. As shown in Table 3, our Edge32 **w/** L method has the shortest training time and relatively high quality.

Table 3: Comparison of our logistic distribution-based sampling method with other methods.

| | NeuS | | NeuS **w/** Hash | | Edge32 **w/** L | |
|---|---|---|---|---|---|---|
| | Chamfer | Time | Chamfer | Time | Chamfer | Time |
| Scan65 | 0.95 | 13072 | **0.78** | 8970 | 0.87 | **7183** |
| Scan106 | 0.75 | 14401 | 0.78 | 7819 | **0.63** | **6476** |
| Scan114 | 0.45 | 15624 | **0.44** | 9600 | **0.44** | **7482** |

**Error sampling result.** In Figure 5, we present our RGB error-based importance ray sampling method (ES) which enhance the image synthesis ability. The light source in the upper left corner is a difficult feature to synthesis. Compared with other methods, our ES method has a leading result.

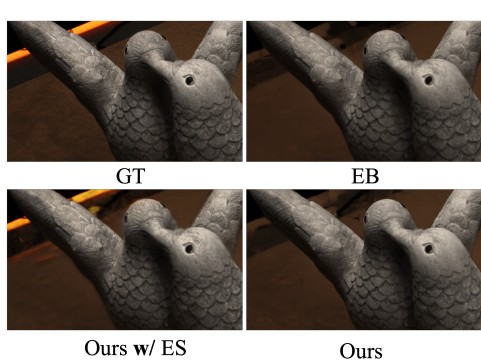

GT                EB

Ours **w/** ES        Ours

Figure 5: Comparisons of using error bounded (EB), RGB error importance sampling (ES), and our edge-sampler approach on DTU Scan106. Compared with other methods, ours can better recover the shape details while using significantly less training time.

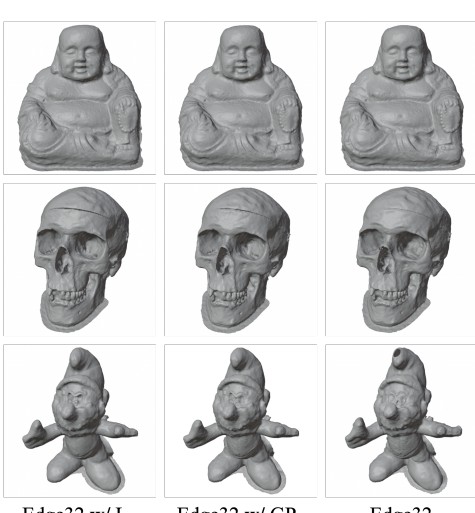

Edge32 **w/** L    Edge32 **w/** GP    Edge32

Figure 6: Comparisons of our three fitting methods, linear interpolation (L), Gaussian Process (GP), and no interpolation. The linear interpolation method produces the best results numerically and visually.

### 5.3 ABLATION STUDY

In addition, we conduct ablation an ablation study on the fitting strategies in Figure 6, which verifies the choice of linear interpolation in our full model. Notably, linear interpolation does not only produce the best results but also runs faster than the Gaussian Process method.

## 6 CONCLUSION

In general, we proposed an efficient and accurate sampling method dedicated to neural implicit surfaces based on Laplace distribution and logistic distributions. The key contributions include an efficient double-sampling strategy to search the tight surface edge, a more accurate integrated weight bound for distributing weight samples, and the evaluations of various fitting approaches. All these technologies come together to increase the training speed by 2x to 3x, and further achieve the SOTA geometry reconstruction quality among existing neural implicit surfaces methods.

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
