# OpenReview forum: "Edge-Sampler: Efficient Importance Sampling for Neural Implicit Surfaces Reconstruction"
_ICLR.cc/2024/Conference — Submitted to ICLR 2024_

### Official Review · Reviewer_17Yv · 2023-10-29

**Soundness:** 3 good
**Presentation:** 2 fair
**Contribution:** 2 fair
**Rating:** 3
**Confidence:** 5

**Summary:**

The paper proposes an efficient sampling strategy for neural surface reconstruction with the SDF representation. Specifically, the authors focus on the edge/surface areas and designed a strategy to sample more points in these areas. The motivation is good and the pipeline looks reasonable. E

**Strengths:**

The paper studies the sampling problem in SDF-based neural surface reconstruction and proposes a four-step sampling strategy, which looks reasonable and might be practical.

**Weaknesses:**

- Novelty: efficient sampling has been studied for relatively a long time since NeRF was proposed. From my perspective, it is not exciting to see research on such a topic. I would expect a significant improvement in efficiency/quality if the paper is to be accepted by a top-tier conference.

- Sampling time comparison: I cannot get the key points from Table 1. Which is the proposed method (Edge32 w/ GP)? Looks like the original NeuS sampling is already the best.

- Comparison with NGP: from Table 3 it looks like NGP is already fast and with high quality. This is also my major concern, different representations (MLP, NGP, K-planes, 3D Gaussians) would have very different sampling strategies, making the method hard or even impossible (e.g. 3D Gaussians) to apply to those representations. Also, powerful new representations would weaken the contribution (efficiency) of the paper.

- The authors only conduct experiments on 4 DTU scenes, which is not convincing to me.

- Too many equations in the paper. I would further simplify equations 1-8 as it is only a recap of previous works.

- The writing of the paper could be further improved.

**Questions:**

See above.

---

### Official Review · Reviewer_qAAj · 2023-10-29

**Soundness:** 3 good
**Presentation:** 3 good
**Contribution:** 3 good
**Rating:** 5
**Confidence:** 2

**Summary:**

This paper presents a new sampling technique called Edge-Sampler for efficient training of neural implicit surfaces in 3D reconstruction. Neural implicit surfaces, which use signed distance functions (SDFs), have shown improved geometry reconstruction compared to neural radiance fields (NeRFs). However, training SDFs is challenging and time-consuming due to the need for large sample counts to capture thin surface edges. The existing error-bounded sampling technique is effective but requires a large number of samples. The Edge-Sampler technique reduces the required sample counts by up to 10x while still preserving the error bound by reducing Riemann integral bias. It achieves this by using a double-sampling strategy to detect thin intervals of surface edges and fitting density functions with bounded CDF errors. The technique demonstrates superior results in terms of geometry reconstruction details, reduced sample counts and training time, and generalizability to different implicit SDF frameworks.

**Strengths:**

-A fast, accurate, and generalized sampling method, namely Edge-Sampler is proposed for high fidelity neural implicit surfaces models training acceleration.
-Several training strategies, including a double-sampling strategy, piecewise fitting and Gaussian Process strategies, and an error-guided ray sampling strategy have been proposed.
-Experimental results show the advantages of the proposed method both in the Sampling time and the accuracy of geometry reconstruction.

**Weaknesses:**

-It seems that the ablation study of the proposed method is not sufficient. Only different fitting strategies are shown in Figure 6. More ablation studies related to other strategies are needed.
-It seems that there is no captions for Figure 4, and there is also not any explanations in the main part of the paper for this figure.

**Questions:**

-It seems that the ablation study of the proposed method is not sufficient. Only different fitting strategies are shown in Figure 6. More ablation studies related to other strategies are needed.
-It seems that there is no captions for Figure 4, and there is also not any explanations in the main part of the paper for this figure.

---

### Official Review · Reviewer_ZFpx · 2023-10-29

**Soundness:** 1 poor
**Presentation:** 2 fair
**Contribution:** 1 poor
**Rating:** 1
**Confidence:** 5

**Summary:**

The manuscript introduces an efficient pipeline for the reconstruction of surfaces from images. Using an SDF-based rendering pipeline such as VolSDF or NeuS, it proposes to narrow the sampling regions in space for more efficient reconstruction. The manuscript has very limited results on DTU dataset.

**Strengths:**

-	The manuscript is easy to follow, with all concepts clearly explained.

**Weaknesses:**

-	Theoretical limitation

  -	The proposed approach assumes that narrower sampling leads to more efficient surface reconstruction. However, this assumption doesn't hold when the optimization process begins with an unknown underlying surface. This might result in slower convergence, particularly when the initialization is far from optimal. The manuscript does not currently acknowledge or address this limitation.

-	Incomplete manuscript
  -	The evaluation of reconstruction accuracy is based on only four scenes from the DTU dataset. The selected results are only for Lambertian surfaces, while many prior works showcase results of specular surfaces of DTU dataset as well. Without insights into the results for the entire dataset, it's challenging to assess the performance comprehensively.
  -	The choice of a single baseline (NeuS) for surface accuracy evaluation is limiting, ignoring the recent developments in surface reconstruction techniques, including but not limited to [Yariv et al, 2021] [Wang et al, 2022] [Fu et al, 2022] [Wang et al, 2023] [Li et al, 2023] [Li et al, 2023] [Li et al, 2023] and etc.
  -	While the manuscript claims improved efficiency, the speed comparison neglects the Occupancy Grid technique proposed in Instant NGP [Muller et al, 2022]. Furthermore, the proposed approach is slower than NeuS unless NeuS' interpolation technique is employed. Moreover, an important baseline is missing, which is the combination of the proposed network and NeuS.
  -	Figure 4 doesn’t have captions.
  -	Table 3 time doesn’t have units.

-	Mismatch motivation and implementation
  -	The manuscript contains a significant discrepancy between its stated motivation and its actual implementation. In the introduction, the manuscript emphasizes the advantages of the Laplace distribution as seen in VolSDF [Yariv et al, 2021] compared to other SDF-based rendering methods. However, in the methodology section, the proposed sampling approach appears to be predominantly focused on enhancing VolSDF. Surprisingly, in all the experiments, it appears that the approach utilizes NeuS' formulation instead. It is important to justify if this is the case and why so.


References:

Yariv, L., Gu, J., Kasten, Y., & Lipman, Y. (2021). Volume rendering of neural implicit surfaces. Advances in Neural Information Processing Systems, 34, 4805-4815.

Wang, Y., Skorokhodov, I., & Wonka, P. (2022). Hf-neus: Improved surface reconstruction using high-frequency details. Advances in Neural Information Processing Systems, 35, 1966-1978.

Müller, T., Evans, A., Schied, C., & Keller, A. (2022). Instant neural graphics primitives with a multiresolution hash encoding. ACM Transactions on Graphics (ToG), 41(4), 1-15.

Fu, Q., Xu, Q., Ong, Y. S., & Tao, W. (2022). Geo-neus: Geometry-consistent neural implicit surfaces learning for multi-view reconstruction. Advances in Neural Information Processing Systems, 35, 3403-3416.

Wang, Y., Han, Q., Habermann, M., Daniilidis, K., Theobalt, C., & Liu, L. (2023). Neus2: Fast learning of neural implicit surfaces for multi-view reconstruction. In Proceedings of the IEEE/CVF International Conference on Computer Vision (pp. 3295-3306).

Li, Z., Müller, T., Evans, A., Taylor, R. H., Unberath, M., Liu, M. Y., & Lin, C. H. (2023). Neuralangelo: High-Fidelity Neural Surface Reconstruction. In Proceedings of the IEEE/CVF Conference on Computer Vision and Pattern Recognition (pp. 8456-8465).

**Questions:**

-	Can the authors discuss if the assumption of a narrow sampling region is valid during the start of optimization?
-	Can the authors justify why only limited results are shown?
-	Can the authors clarify if the approach uses NeuS’ density conversion instead of VolSDF?

---

### Official Review · Reviewer_Xm98 · 2023-10-30

**Soundness:** 3 good
**Presentation:** 2 fair
**Contribution:** 2 fair
**Rating:** 5
**Confidence:** 3

**Summary:**

This paper indicates the time-consuming issue caused by the large sample counts on sign distance function (SDF) thin edges in the field of neural implicit surfaces while employing error-bounded sampling. To resolve the issue, the authors propose a four-step edge sampler to reduce the required sample counts while preserving the theoretical error bound. Precisely, the double-sampling step detects the tiny edge interval to reduce the calculation cost. The error-bounded weight sampling step fits a PDF under a bounded error. The sample interpolation estimates the Riemann sum error for any uniform sample. The last error sampling is used to enhance the image synthesis ability. The experiment shows the efficiency of the proposed edge sampler on the DTU dataset.

**Strengths:**

-	The paper is well-motivated, which indicates the time-consuming issue caused by the large sample counts on sign distance function (SDF) thin edges while employing error-bounded sampling.

**Weaknesses:**

The method description shows many details. It is better to attach an algorithm to summarize the entire sampling mechanism.

The experiments need to demonstrate the advantage of the proposed four-step edge sampler clearly.
* The model configurations of the compared methods in section 5.1 need to be clarified.
* The DTU dataset comprises many scenes, and it is better to include the complete results for performance comparison.
* It lacks a table to compare the reducing sample counts against other methods.
* Figure 4 has no caption.
* What is the unit of time in Tables 2 and 3? Is it fair to compare time using the batch size more than one?
* The visualization differences in Figures 4 and 6 are hard to perceive.
* What causes the performance degradation of NeuS in Table 3 compared to their paper?

**Questions:**

The primary concern of this paper is its weak experiments. It is better to strengthen the experiments as detailed in the [Weaknesses]; otherwise, the reviewer does not know that the comparison could fairly reflect the performance improvement by the proposed method.

---

### Meta-Review · Area_Chair_UnmS · 2023-12-07

**Metareview:**

The submission received negative reviews from all the reviewers. The reviewers' main concerns were mostly around the clarity of the method and unpolished presentation, as well as insufficient experiments. The authors did not submit a rebuttal. After reading the paper and the reviewers' comments, the AC agrees with the decision by the reviewers and recommends rejection.

**Justification For Why Not Higher Score:**

The reviews are inclined towards rejection, with the most critical part being the clarity of the method, unpolished presentation, and insufficient experiments.

**Justification For Why Not Lower Score:**

N/A

---

### Decision · Program_Chairs · 2024-01-16

Reject